# Effectiveness of therapeutic ultrasound for the treatment of carpal tunnel syndrome (the USTINCTS trial): study protocol for a three-arm, prospective, multicentre, randomised controlled trial

Shuai Chen [ID],[1,2] Yun Qian [ID],[1,2] Ziyang Sun [ID],[1,2] Weixuan Liu,[1,2] Guixin Sun,[3] Junjian Liu [ID],[4] Jian Wang,[5] Wei Wang,[1,2] Yuanyi Zheng,[6] Cunyi Fan[1,2]

SC, YQ and ZS contributed equally.

For numbered affiliations see end of article.

**Correspondence to**
Professor Cunyi Fan;
cyfan@sjtu.edu.cn and
Professor Yuanyi Zheng;
zhengyuanyi@163.com

## ABSTRACT

**Introduction** There has no consensus on optimal management of carpal tunnel syndrome (CTS), the most common compression neuropathy. Conservative therapy is generally accepted as first-line intervention. Therapeutic ultrasound has been widely reported to be treatment beneficial in nerve regeneration and conduction, and further accelerate compression recovery. The purpose of this study is to investigate the effectiveness of therapeutic ultrasound for CTS treatment.

**Methods and analysis** This study protocol entails a three-arm, prospective, multicentre, randomised controlled trial. 162 eligible adult participants diagnosed with mild to moderate CTS by using criteria developed from a consensus survey by the UK Primary Care Rheumatology Society will be assigned to either (1) therapeutic ultrasound, (2) night splint or (3) therapeutic ultrasound +night splint (combined) group. Primary outcome will be difference in Symptom Severity Scale of Boston Carpal Tunnel Questionnaire (BCTQ-SSS) at 6-week between night splint and therapeutic ultrasound +night splint groups. Secondary outcomes include Functional Status Scale of BCTQ, sleep questionnaire for interrupted sleep, EuroQol-5D for general health, Hospital Anxiety and Depression Scale for mental status, Work Limitations Questionnaire-25 for functional limitations at work, Global Rating of Change for treatment success and recurrence rate, physical examination, electrophysiological and ultrasound parameters. Intention-to-treat analyses will be used.

**Ethics and dissemination** Ethics committees of all clinical centres have approved this study. The leading centre is Shanghai Sixth People's Hospital, whose approval number is 2021-152. New versions with appropriate amendments will be submitted to the committee for further approval. Final results will be published in peer-reviewed journals and presented at local, national and international conferences.

**Trial registration number** ChiCTR2100050701.

## Strengths and limitations of this study

► Therapeutic ultrasound as independent or adjunct therapy in treating carpal tunnel syndrome (CTS).
► The first randomised controlled trial (RCT) to compared the efficacy between therapeutic ultrasound and night splint in CTS treatment.
► Multicentre RCT with blinded outcome assessor and statistician.
► Use of several patient-reported outcome measures as well as objective parameters.
► Participants and treating surgeons not blinded.

from median nerve entrapment in the carpal tunnel, accounting for about 90% of all such disorders.[1 2] The clinically confirmed CTS prevalence was 9.6% in the general population of China,[3] with a yearly incidence rate of 2.76‰, and women is more susceptible than men.[2 4] CTS has significant impact on daily life and ability to work,[5] and causes great burden on social economy, with an annual associated costs estimated at US$13 billion.[6] Classically, CTS causes discomfort, paraesthesia and numbness in the median nerve distribution; and nocturnal symptoms are often clinically significant causing sleep disturbance.[7] Patients can be diagnosed by clinical history and physical examination; while electrophysiological methods will be additional for insufficient diagnosis by clinical findings and severe cases that need surgical management.[8]

In general, the severity of CTS can be classified into mild, moderate and severe.[9] Non-surgical interventions are suggested to be the first choice to treat mild and moderate CTS.[10] To date, though the treatment method is vast; however, no successful and universally accepted regimen has been established. A consensus of multidisciplinary treatment

## INTRODUCTION

Carpal tunnel syndrome (CTS), the most common compression neuropathy, results

guideline from the European HANDGUIDE Study suggests that 'education' should be included as the first-line management approach, which has the advantages of low cost, high efficacy and non-invasiveness.[7 11] In addition, 'night splint' and 'corticosteroid injection' are also recommended in guidelines of American Academy of Orthopaedic Surgeons and American Physical Therapy Association.[12 13] One recent randomised controlled trial (RCT) published in Lancet compares both two methods, and finds that corticosteroid injection has superior clinical effectiveness at 6 weeks than night splint, but no differences at 6 months; while corticosteroid injection may bring adverse events like thinning, lightening or darkening of the skin at the injection site, hot flushes and even more pain.[9] Systematic reviews have also shown that the effects of other conservative treatments like acupuncture,[14] exercise and mobilisation interventions,[15] laser,[16] extracorporeal shockwave therapy[17] and platelet-rich plasma injection[18] still remain controversial or provide little to no benefit.

Therapeutic ultrasound (US) is widely used for imaging purposes and regarded as an adjunct to physiotherapy. In the intensity range of 0.5–2.0 W/cm$^2$, therapeutic US may have the potential to induce a variety of biophysical effects in tissues.[19] Therapeutic US experiments on stimulation of nerve conduction and regeneration,[20 21] and discoveries of its anti-inflammatory effects[22] all support that therapeutic US may promote recovery of nerve compression. An RCT published in BMJ showed more pronounced subjective symptoms and electroneurographic variables for therapeutic US than sham control in patients with mild to moderate CTS.[23] However, to our best of knowledge, no study has compared the efficacy between night splint and therapeutic US in CTS treatment yet. Additionally, some studies have also reported the efficacy of US to be used as part of a multi-intervention approach, but with low grade of study design and data.[24–29] Therefore, the role of therapeutic US in CTS treatment still needs to be further explored by high-quality study.

Therefore, the purpose of the current three-arm, prospective, randomised, multicentre trial is to examine the effectiveness of therapeutic US in treatment for CTS, that is, night splint +therapeutic US (combined) vs night splint vs therapeutic US, on clinical and functional outcomes, including Boston Carpal Tunnel Questionnaire (BCTQ) in patients diagnosed with CTS.

## METHODS
### Study design
This study is a three-arm, prospective, multicentre, RCT that will recruit participants from four municipal tertiary hospitals with a diagnosis of mild to moderate CTS. The multicentres are Shanghai Sixth People's Hospital, Shanghai Tenth People's Hospital, Shanghai East Hospital and Pudong New Area People's Hospital of Shanghai, respectively. This manuscript is written according to the

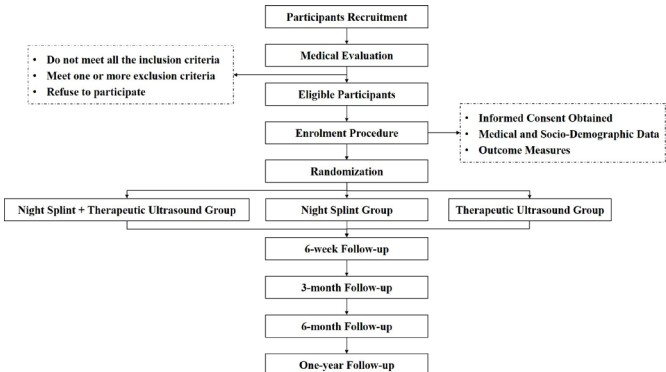

**Figure 1** Participant flow chart.

Standard Protocol Items: Recommendations for Interventional Trials (SPIRIT) guidelines.[30]

### Participant and public involvement
There was no participant involvement in this study. Participants were not invited to make comments or suggestions on the protocol, were not consulted about the selection of patient-relevant evaluation outcomes, or were invited to participate in writing or editing the manuscript to improve readability or accuracy. The final findings will be disseminated to the public through mass media. Our published papers and conference presentations also acknowledge all participants as a whole.

### Participant recruitment
Figure 1 shows the participant flow chart throughout the study. Participants will be recruited over a 5-month period from outpatient clinics of four principals in each subcentre. In addition, recruitment can also be through other doctors and healthcare professionals. Interested participants can contact the research assistant, who will provide more information about the study purpose and protocol, and conduct an initial eligibility screening by phone.

### Medical evaluation and enrolment procedure
Eligible participants will be invited to participate in a physical examination to confirm CTS diagnosis and assess eligibility to participate in the programme.

#### Inclusion criteria
► Age ≥18 years old.
► Clinical diagnosis on the basis of clinical symptoms, history and physical examination, using criteria developed from a consensus survey by the UK Primary Care Rheumatology Society.[31]
► Mild to moderate CTS,[9] with symptoms longer than 6 weeks duration; participants with bilateral CTS will be designated their study hand based on the most severe symptoms.
  – Mild: intermittent paraesthesia in the distribution of the median nerve.

- – Moderate: constant paraesthesia, and reversible numbness or pain of idiopathic nature.
  - – Severe: constant pain, numbness or sensory loss in the wrist and hand (specifically palm, index, or middle finger, or thumb), or thenar muscle atrophy.
► Able to read and write in simplified Chinese (Mainland), understand and complete the questionnaire, and should provide informed consent.

## Exclusion criteria

► CTS secondary to wrist deformity, trauma, mass, pregnancy, hypothyroidism or inflammatory arthropathy.
► Treatment by night splint, therapeutic US or injection within the past 6 months or previous carpal tunnel surgery.
► Previous surgery on the affected wrist (or study wrist if bilateral CTS).
► Unable to tolerate the study interventions.
► Trauma to the affected upper limb requiring operation or immobilisation within the past 12 months.
► Current illness including, poorly controlled diabetes mellitus or thyroid disease, vibration-induced neuropathy, osteoarthritis or inflammatory joint disease, suspected complex neurological and musculoskeletal conditions.
► Known drugs or alcohol abuse.
► Allergy to any of the splint materials.
► Contraindications to therapeutic US, including dermatological conditions, abnormal sensation in the affected arm, indwelling electrical pumps/pacemakers, epilepsy, pregnancy or breast feeding.

Following the medical evaluation, a research assistant will meet with the eligible participants and obtain their written informed consent. Demographic variables such as age, sex, body mass index, affected wrist (whether bilateral), dominant arm, lifestyle (smoking and alcohol use) and medical history of all participants will be collected prior to treatment (baseline). Participants will also be asked relevant questions about symptoms duration and previous treatments (rehabilitation exercises, injections or others). Others like occupation, employment characteristics (full-time or part-time work, manual or nonmanual labour), employment status (whether on sickness absence), and professional activity characteristics (repetitive movements for >4 hours/day; wrist flexion and extension for >2 hours/day and use of computer keyboard/mouse (how many hours/day)) will be also collected.

## Randomisation and blinding

Participants will be randomised in three intervention groups (either therapeutic US or night splint or therapeutic US+night splint arm) in a ratio of 1:1:1, using a computer-generated randomised sequence with varying unknown block sizes (either 3 or 6) for all study centres, without stratification. A research assistant who are not involved in clinical care and participant evaluations will prepare sequentially numbered, opaque, sealed envelopes based on a random list, and ensure that the allocation data will not be accessed or influenced by anyone. When appropriate, the assistant will open envelopes and ensure coordination of therapeutic interventions.

The outcome assessor and statistician will be blinded to group allocation and not involved in treatment procedures.

## Intervention

At the beginning, all participants will participate in an about 30 min group educational presentation by a research assistant on the same day as the baseline assessment. This presentation will cover the pathophysiology, treatment options, posture and activity modification principles of CTS. The above information will also be provided to participants in the form of education booklets, to encourage them to review at home. Habits changes include limited wrist movement and a reduction in strenuous work activities, and the use of ergonomically friendly work tools helps reduce median nerve pressure.[32 33]

Participants in the (therapeutic US group) will receive pulsed therapeutic US (model 1:4, Shanghai, China) for 6 weeks at a frequency of 1 MHz and intensity of 1.0 W/cm$^2$ for 15 min per session, in daily five times a week for the first 2 weeks and twice a week for another 4 weeks, to the area over the carpal tunnel, referred to a published trial (BMJ. 1998;316:731–735).

Participants allocated to the (night splint group) will receive a splint to wear at night for 6 weeks, referred to a published trial (Lancet. 2018;392:1423–1433). The splint holds the wrist in a neutral position or slightly extended 20° from the neutral position to avoid wrist movement, which can increase pressure on the carpal tunnel.[34] The choose of each splint will be based on the size of individual participant's hands and arms. Participants will be shown how to fit and remove the wrist splint according to a standardised trial protocol. Oral guidance from the clinician on how and when to use splints will encourage and reinforce compliance, which will be also supported by written information, detailed care and splint fitting and use. Participants will be instructed to perform gentle range-of-motion exercises when removing the splint to prevent stiffness.

Participants randomised to the (therapeutic US+night splint group) will receive both therapeutic US for 6 weeks as in the (therapeutic US group) and night splint for 6 weeks as in the (night splint group).

For participants with bilateral CTS, the non-study hand will be treated according to normal clinical protocols in use at the research site.

We discouraged additional treatments to that assigned (ie, not per protocol) during the intervention period, but we allowed the use of simple analgesics as needed. Participants reported all not per protocol treatments, such as drugs, in a diary.

**Table 1** Study evaluation procedures and timeline

| Study procedure | Medical evaluation | Enrolment visit | 6 weeks | 3 months | 6 months | 1 year |
|---|---|---|---|---|---|---|
| Determine eligibility | √ | √ | | | | |
| Obtain signed consent | | √ | | | | |
| Obtain medical and demographic data | | √ | | | | |
| Give instructions for pain medication diary | | √ | | | | |
| Outcome measures | | | | | | |
| Boston Carpal Tunnel Questionnaire | | √ | √ | √ | √ | √ |
| Interrupted Sleep Questionnaire | | √ | √ | √ | √ | √ |
| European Quality of Life Scale (EuroQol)-5D | | √ | √ | √ | √ | √ |
| Hospital Anxiety and Depression Scale | | √ | √ | √ | √ | √ |
| Work Limitations Questionnaire-25 | | √ | √ | √ | √ | √ |
| Treatment success rate | | | √ | √ | √ | √ |
| Treatment recurrence rate | | | | √ | √ | √ |
| Physical examination | | √ | √ | √ | √ | √ |
| Electrophysiological and ultrasound parameters | | √ | √ | √ | √ | √ |

## Data management

Data will be collected during the participants' visits to the hospital at baseline, 6 weeks, 3 months, 6 months and 1 year after random assignment (table 1). Reminder emails and phone calls from the research assistants will be programmed to maximise participant compliance in subsequent completion. A registered participant will withdraw from the study if (1) the participant withdraws his/her consent and (2) exclusion criteria is found after registration. The cause and date of suspension will be recorded. Consent to use data that has been collected before the participant's withdrawal will be included in the consent form.

## Primary outcome measure

The primary outcome measure will be the difference in Symptom Severity Scale of the Boston Carpal Tunnel Questionnaire (BCTQ-SSS) at 6 weeks. The BCTQ is a disease-specific questionnaire referring to a typical 24-hour period in the past 2 weeks,[35] and has been shown to be highly reproducible, internally consistent, valid and responsive to clinical change in CTS.[36] It consists of two different subscales: Symptom Severity Scale (11 items, about symptom severity) and Functional Status Scale (8 items, about the degrees of difficulty on daily activities), both rated on a five-point scale, with final scores for each subscale result in mean scores between 1 and 5. The overall score is calculated as the mean of all 19 items. Higher scores represent more severe symptoms

and functional impairment. We use a validated Chinese version[37] of the BCTQ in this study.

## Secondary outcome

Secondary outcome measures will be the differences in Functional Status Scale of the Boston Carpal Tunnel Questionnaire (BCTQ-FSS), sleep questionnaire for interrupted sleep,[38] European Quality of Life Scale (EuroQol)-5D (EQ-5D)[39] for life quality and health status, Hospital Anxiety and Depression Scale (HADS)[40] for anxiety and depression status, Work Limitations Questionnaire (WLQ)-25[41] for functional limitations at work, Global Rating of Change (GROC) for treatment success and recurrence rate, physical function examination as well as various electrophysiology and US parameters.

► Interrupted sleep
  The sleep questionnaire, which will be used to assess sleep quality, consists of four questions asking participants how many times they have experienced it in the last month.[38] Each question has six response categories and are coded in 0–5 order: not at all, 1–3, 4–7, 8–14, 15–21 and 22–31 days. All questions have equal weights and add up. Higher scores are associated with more disrupted sleep.
► Life quality and health status
  EQ-5D has been widely validated and used to measure generic health-related quality of life (HRQol) due to its simplicity.[39] It consists of a five-dimensional description system (mobility, self-care, usual activities, pain/

discomfort and anxiety/depression) and a visual analogue scale that scales from 0 to 1, where 1 represents perfect health. All dimensions are divided into three levels (no problem, some problem and extreme problem). We used a validated Chinese version[42 43] of EQ-5D, which has been recommended by Guidelines for Pharmacoeconomic Evaluations 2011 for a measure for HRQol and health utility.[44]

► Anxiety and depression status
HADS will be used to identify and quantify two of the most common psychological disorders - anxiety and depression.[40] There is evidence of increased anxiety and depression in LET patients.[45] The HADS is a 14-item scale independent of physical symptoms and consists of two 7-item subscales measuring depression and anxiety, respectively. A four-point scale (0 for absence of symptoms and 3 for maximum symptomatology) is used. Each subscale has an overall score ranging from 0 to 21, with a higher score indicating a higher degree of impairment. HADS has two cut offs for categorisation: 0–7, 'non-case'; 8–10, 'possible or doubtful case'; 11–21, 'probable or definite case'.[46]

► Functional limitations at work
To gather information that is complementary to the pain and disability scales, functional limitations at work will be measured using WLQ-25. It contains 25 items, arranged under four subscales, covering four dimensions of job demands, namely time demands, physical demands, mental/interpersonal demands and output demands.[41] A five-level ordinal response scale ranging from 0 (all of the time) to 4 (none of the time) with an additional sixth option (does not apply to my job) is used. The overall score ranges from 0 to 100, with an increase of 13 points (out of 100) for clinically important differences.[47]

► Treatment success and recurrence rate
Participants' treatment impressions of changes in their condition (ranging from 'completely recovered', 'much improved', 'somewhat improved', 'same', 'worse' to 'much worse') will be recorded on a six-level Likert scale. The success rate will be calculated by dichotomising the response. Participants who report 'completely recovered' or 'much improved' in their overall condition since the study beginning will be considered successful, while other responses will be considered failures.[48 49] Recurrence will be defined primarily as when a participant rates a success at 6 weeks and a failure at 3 months, 6 months or 1 year on GROC.[48 49]

► Physical function examination
The physical examinations will include measurement of two-point discrimination (performed on the radial and ulnar aspects of each digit), grip strength with a dynamometer (CAMRY, City of Industry, California, USA), and pinch strength with the pinch gauge (three trials for each hand). The affected side will be measured first and then the unaffected side. The measurement readings will be not revealed to the subjects until the completion of the test. The mean of three consecutive trials, separated by a 20 s pause, will be calculated. Results will be presented as a ratio of values of the symptomatic side/asymptomatic side ×100.[50]

► The two-point discrimination test starts at a distance of 4 mm and increases continuously by 2 mm as necessary. Grip strength and three-point pinching force (three tests per hand) as measured with baseline dynamometer and pinch gauge (Chattanooga Group, Hixson, Tennessee, USA) will be recorded.

► Electrophysiological study
Median nerve distal motor latency (DML), compound muscle action potential (CMAP), sensory nerve conduction velocity (SNCV) and sensory nerve action potential (SNAP) amplitudes will be recorded.[48 49] DML and CMAP will be measured by placing surface electrodes on the abductor pollicis brevis muscle, and stimulation applied 8 cm proximal to the active recording electrode. SNAP and SNCV will be obtained using ring electrodes placed on the proximal and distal interphalangeal joints of the index finger. Sensory conduction will be studied by antidromically stimulation at 14 cm proximal to the active electrode. Motor study will be performed by supramaximal stimulation while the amplitude will be measured an average of 10 times for the sensory study. All measurements will be made three times, and the values obtained will be averaged for analysis.

► US parameters
The cross-sectional area will be measured using an electronic calliper at the scaphoid-pisiform level.[51 52] The measurements will be made three times, and the values obtained will be averaged for analysis.

## Adverse events

All adverse events, defined as any negative or unwanted reactions to intervention, will be recorded through the symptoms reported by the patients, and observations by a researcher at every visit. Therapeutic US may cause mild local swelling, spot-like bleeding, ecchymosis, enhanced local pain response and local hyperesthesia or decrease. Night splint may cause skin allergy, wrist stiffness, etc. The participants will be instructed to do gentle range-of-motion exercises when removing the splint to prevent stiffness and reinforced adherence by verbal instruction.

## Sample size calculation

Sample size and power calculation are based on the primary outcome of BCTQ-SSS score at 6 weeks. All sample size calculations assume two-sided analysis with a power of 90% (1-β=0.90) at a significant level of α=0.05. Based on a published RCT trial (Lancet. 2018;392:1423–1433), the authors compared corticosteroid injection and night splint for CTS, and participants in the night splint group had an SD (SD) of 0.76 points for BCTQ-SSS at their 6-week follow-up, ('6 weeks' was the primary endpoint in this study)),[9] an SD of 0.76-point on BCTQ-SSS score will be used. To detect a minimum clinically significant

difference of 0.8-point[53] (superiority margin) between therapeutic US+night splint and night splint groups (assuming a true difference of 1.19-point,[9 54] a total of 48 participants in each group is required. Allowing for an up to 10% drop-out rate, we aim to enrol at least 54 participants in each group to complete the study.

## Analysis plan

Baseline characteristics of the three treatment groups will be summarised using appropriate descriptive statistics. Both the primary and secondary analyses will be blind analyses of treatment assignments and will be performed using the intention-to-treat[55] method, with all randomised participants retaining their original assigned group. Multiple imputation by chained equations will be used to address missing data caused by lost access and non-response if these missing data are judged to be random.

The primary comparisons for BCTQ-SSS scores will be made using linear regression. In secondary analyses, repeated measures mixed model[56] will also be used to examine the associations between treatments and repeated outcome measures, with terms of treatment, time, trial centre and corresponding baseline values as covariates (age, gender, body mass index, etc). Bonferroni method will be used to adjust for multiplicity.[57 58] Linear regression will be used for numerical outcomes, and logistic/ordinal regression for any categorical outcomes.

## Quality assurance/monitoring/management

In order to standardise the procedures of staff training and learning, such as participants recruitment, outcome measures, data import, security, management and analysis, a manual of operations and procedures and a case report form will be developed as per protocol, which also include the monitoring plans to assure participant protection and data integrity, thus facilitating consistency in protocol implementation and data collection. The investigators, physicians, research assistants, outcome assessors and statisticians are different people and should be trained in good clinical practice. Trained project managers will visit each centre for monitoring to ensure data quality and compliance with the trial protocol.

All data obtained will be stored electronically and strictly in a database with secure and restricted access. Encryption will be used for data transmission, with removal for any information that can identify individuals. Data will only be deidentified for analysis at the completion of this study.

## Study duration

Recruitment of the trial will begin in the November of 2021 and 1-year follow-up for all participants is anticipated to be completed by June 2023. See table 1 for time points and recruitment progress.

## ETHICS AND DISSEMINATION

This study has been approved by the Ethics Committee of Shanghai Sixth People's Hospital (lead Clinical Center,

approval No. 2021-152), Ethics Committee of Shanghai East Hospital (EC.D(BG).016.03.1-2021-095), Ethics Committee of Shanghai Tenth People's Hospital (SHSY-IEC-4.1/21-194/01) and Ethics Committee of Pudong New Area People's Hospital (IRBY2021-006). The potential risks of this clinical trial are considered minimal and are addressed in the protocol and consent form. A written consent (online supplemental file 1) will be obtained by clinical practitioners from each participant. Data will be published in peer-reviewed journals and presented at conferences, both nationally and internationally.

## DISCUSSION

CTS is a highly prevalent compression neuropathy, which results in significant paraesthesia and numbness in the median nerve distribution, especially nocturnal symptoms causing sleep disturbance, causing great socioeconomic burden. Up till now, there is still no consensus on the optimal management, and non-operative treatment is generally accepted as the first-line intervention for mild and moderate CTS. Multiple methods have been studied and reviewed in the recent decades, however, the exact efficacy still remains controversial.

In an RCT published in BMJ for mild to moderate CTS, active therapeutic US (1 MHz, 1.0 W/cm$^2$) was applied to the area over the carpal tunnel in the experimental group, and indistinguishable sham US treatment was applied in the control group.[23] At 6 months' follow-up, satisfactory improvement or complete remission of symptoms was observed in 74% receiving active treatment, which is significantly higher than those receiving sham treatment (20%). As for electroneurography, DML and SNCV improved significantly with active treatment while remained unchanged with sham treatment. Hand grip and finger pinch strength in physical examination also improved significantly with active treatment. All results suggested satisfying effects from therapeutic US for CTS.

Therapeutic US can also be used as part of a multi-intervention approach. Some studies have compared night splint alone to night splint combined with therapeutic US in treatment of CTS, while the effects were different, and the grades of study design and data were low. Dincer et al[26] found that the improvements in the combined group were statistically significantly better (p=0.043) than those in night splint alone group in BCTQ-SSS, as well as BCTQ-FSS (p<0.001), and VAS for pain (p<0.001). Similar results were also reported by Baysal et al,[59] while Jothi and Bland,[27] Sim et al[28] and Armagan et al[29] found therapeutic US may add no benefit to splinting in CTS.

In this study, to the best of our knowledge, it is the first to compared the efficacy between therapeutic US (therapeutic US group) and night splint (night splint group) in CTS treatment. What's more, the additional effects of therapeutic US (therapeutic US+night splint group) in a multi-intervention approach will be compared with night splint alone (night splint group). In clinic, therapeutic US is less invasive, less expensive, safer and more portable than other nonoperative

therapy like drug injections for compression neuropathy and, if proved to be effective, could be offered to selected patients as part of non-operative therapy.

There are some ongoing clinical trials on CTS treatment recent years,[60–63] and our prospective randomised study proposes to complement and add to this relevant and much needed scientific effort.

**Author affiliations**
[1]Department of Orthopedics, Shanghai Jiao Tong University Affiliated Sixth People's Hospital, Shanghai, China
[2]Shanghai Engineering Research Center for Orthopaedic Material Innovation and Tissue Regeneration, Shanghai, China
[3]Department of Orthopaedics, Shanghai East Hospital, Tongji University School of Medicine, Shanghai, China
[4]Department of Orthopedics, Shanghai Tenth People's Hospital, School of Medicine, Tongji University, Shanghai, China
[5]Department of Orthopaedics, Pudong New Area People's Hospital, Shanghai, China
[6]Department of Ultrasound in Medicine, Shanghai Jiao Tong University Affiliated Sixth People's Hospital, Shanghai, China

**Acknowledgements** We will appreciate the support from Base for Interdisciplinary Innovative Talent Training, Shanghai Jiao Tong University and Youth Science and Technology Innovation Studio of Shanghai Jiao Tong University School of Medicine.

**Contributors** SC, YQ and ZS are the primary investigators. SC, YQ, YZ, WL, YZ, CF participated in the development of the study design. SC, YQ, YZ, WL, GS, JL, JW, WW, YZ and CF participated in the study conduct. SC, YQ, ZS and WL drafted the manuscript under CF's supervision. CF and YQ contributed to applying for and gaining funding. All authors contributed to the content and critical revision and approved the final draft of the manuscript.

**Funding** This study will be supported by National Natural Science Foundation of China (8217090787, 82002290); Shanghai Engineering Technology Research Center and Professional Technology Service Platform project of 2020 'Science and Technology Innovation Action Plan' of Shanghai (20DZ2254100); Municipal Hospital Clinical Skills and Innovation Capacity of Three-year Action Plan Programme of Shanghai Shenkang Hospital Development Center (SHDC2020CR2039B, SHDC2020CR6019-002); Biomedical Technology Support Special Project of Shanghai 'Science and Technology Innovation Action Plan' (20S31900300, 21S31902300); Shanghai Sailing Program (No. 20YF1436000); Young Elite Scientist Sponsorship Program by Cast (No. YESS20200153); Clinical Research Center (CRC) of Shanghai University of Medicine and Health Sciences (20MC2020001).

**Competing interests** None declared.

**Patient and public involvement** Patients and/or the public were not involved in the design, or conduct, or reporting, or dissemination plans of this research.

**Patient consent for publication** Consent obtained directly from patient(s)

**Provenance and peer review** Not commissioned; externally peer reviewed.

**ORCID iDs**
Shuai Chen http://orcid.org/0000-0003-3724-0174
Yun Qian http://orcid.org/0000-0003-1600-5693
Ziyang Sun http://orcid.org/0000-0002-8673-9521
Junjian Liu http://orcid.org/0000-0002-3052-3799

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
