## [Reviewer comments · BMJ Open]

ARTICLE DETAILS

TITLE (PROVISIONAL)	Effectiveness of therapeutic ultrasound for the treatment of carpal tunnel syndrome (the USTINCTS trial): study protocol for a three-arm, prospective, multicenter, randomised controlled trial
AUTHORS	Chen, Shuai; Qian, Yun; Sun, Ziyang; Liu, Weixuan; Sun, Guixin; Liu, Junjian; Wang, Jian; Wang, Wei; Zheng, Yuanyi; Fan, Cunyi

VERSION 1 – REVIEW

REVIEWER	Isam Atroshi Hösslholm Hospital, Department of Orthopedics
REVIEW RETURNED	28-Oct-2021

GENERAL COMMENTS	The research question is of interest and is clinically relevant. A treatment that requires patient visits 5 days per week for 6 weeks is not very practical for patients, especially those who have employment, and probably not for most health care providers. It would also be very costly considering that carpal tunnel syndrome is a very common condition. Therefore, for such a treatment method a 3-month outcome would not be of major interest even if the treatment is shown to be relatively effective. To be of interest the benefit should be more durable. The sample size appears to be too small for a 3-arm RCT. The authors base their sample size calculation based on the Boston symptom severity score assuming a standard deviation (SD) of 0.38, which is probably too small. The authors cite a published article supporting their assumed SD but do not mention what the SD in that article refers to, is it SD for follow-up score? After what treatment and after how long follow-up time? There are many studies that have reported SD for the Boston symptom severity score after different treatments, and the authors need to justify why they chose the reported SD in this paper for their sample size calculation. Is the primary outcome the symptom severity score at 3 months or change in score over time? Considering that the trial has three groups and multiple follow-up measurements, there is no mention of how the issue of multiplicity will be managed. The authors chose several secondary outcomes and with this small sample size it would be unlikely that the study will have adequate power for the analysis of these secondary outcomes. Abstract needs to include more information about the eligibility criteria and the primary outcome (timepoint and which groups will be compared).
---

REVIEWER	Philippe Paquette University of Montreal, School of rehabilitation
REVIEW RETURNED	29-Oct-2021

GENERAL COMMENTS

BMJ Open # 2021-057541

Effectiveness of ultrasound therapy for the treatment of carpal tunnel syndrome (the USTINCTS trial): study protocol for a three-arm, prospective, multicenter, randomised controlled trial

This study aims to determine the effectiveness of ultrasound therapy on symptoms and function, sleep, quality of life, patient satisfaction, physiological measures, amongst others on participants diagnosed with mild to moderate carpal tunnel syndrome (CTS). To do so, the authors will prospectively evaluate the effects of: ultrasound therapy; ultrasound therapy + night splints; night splints alone, on clinical and functional outcomes using a three-arm experimental design. The randomization and data management are well thought out and described. This protocol appears relevant for adding robust data to aid clinical decision making regarding the conservative treatment of CTS.

I only have minor comments that the author could consider

- Participant selection and sample size

The sample size estimate appears sound. Given the multicenter scope of this study, would the authors consider increasing the statistical power or perhaps adding a fourth branch to include exercises or sham ultrasound? Additionally, since 21 participants are anticipated in each groups, I am concerned about potential heterogeneity attributed to gender, time since onset, and symptoms severity of CTS.

- Ultrasound therapy parameters

My main concern regards the choice to use continuous therapeutic ultrasound, which can cause overheating in the treatment area. Although the proposed parameters (i.e., 1 MHz 1.0 W/cm²; 15 min; 5 day/week) have been use in prior studies, adverse events have not been thoroughly documented (1). There are safety concerns in the use any type of therapeutic ultrasound which are outlined in Miller et al.(2) Furthermore, the authors should provide stronger arguments in favor of continuous mode, as studies referenced in the introduction highlight that pulsed ultrasound has effects on nerve conduction.

- Secondary outcome measures

The scope of secondary outcomes is unclear. The authors should consider that certain health domains to be evaluated overlap. For instance, functional status is measured in both the BCTQ and Quick-DASH. Likewise, pain and paresthesia are measured in both the BCTQ and with a NRS. Multiple patient reported questionnaires could potentially be burdensome for participants and reduce compliance. Conversely, the authors could consider including an economic or cost-effectiveness evaluation. Additionally, EMG and

1/3

	ultrasound imaging outcomes are relevant, but should be clearly explain. For instance, will there be a single laboratory for EMG studies? Did the author consider other ultrasound imaging measures such as echogenicity and texture? Miscellaneous:  • The abbreviation of Carpal Tunnel Syndrome is CTS. The authors should provide a rational for using CaTS • Please refer to 'therapeutic ultrasound' or 'ultrasound therapy' to avoid confusion with ultrasound imaging and avoid the 'US' abbreviation. References:  1. Page MJ, O'Connor D, Pitt V, Massy-Westropp N. Therapeutic ultrasound for carpal tunnel syndrome. Cochrane Database Syst Rev. 2013(3):CD009601. 2. Miller DL, Smith NB, Bailey MR, Czarnota GJ, Hynynen K, Makin IR, et al. Overview of therapeutic ultrasound applications and safety considerations. J Ultrasound Med. 2012;31(4):623-34.
--	---

VERSION 1 – AUTHOR RESPONSE

□ Reviewer: 1

Dr. Isam Atroshi, H \diamond ssleholm Hospital

The research question is of interest and is clinically relevant.

Comment 1

A treatment that requires patient visits 5 days per week for 6 weeks is not very practical for patients, especially those who have employment, and probably not for most health care providers. It would also be very costly considering that carpal tunnel syndrome is a very common condition. Therefore, for such a treatment method a 3-month outcome would not be of major interest even if the treatment is shown to be relatively effective. To be of interest the benefit should be more durable.

Response

Thanks for your comment! According to your advice, it's actually meaningful to show longer-term outcomings. We have changed the followup timepoint into "6 weeks", "3 months", "6 months", and last to "one year". Thanks for your kind advice again!

In fact, we are developing a portable US machine, which has the same frequency and intensity, so that patients will be able to use it at everywhere, like home and work place. This will reduce the cost and inconvenience of repeated visits. Therefore, if the results of US in this study are promising, we can deliver a further research on the portable US machine in future. Thank you again for your kind advice!

Changed text

Table 1 and Figure 1.

Page 14, Line 291-292.

Page 17, Line 374-375.

Data will be collected during the participants' visits to the hospital at baseline, 6 weeks, 3 months, 6 months, and one-year after random assignment (Table 1).

Recurrence will be defined primarily as when a participant rates a success at 6 weeks and a failure at 3 months, 6 months or one-year on GROC.

Comment 2

The sample size appears to be too small for a 3-arm RCT. The authors base their sample size calculation based on the Boston symptom severity score assuming a standard deviation (SD) of 0.38, which is probably too small. The authors cite a published article supporting their assumed SD but do not mention what the SD in that article refers to, is it SD for follow-up score? After what treatment and after how long follow-up time? There are many studies that have reported SD for the Boston symptom severity score after different treatments, and the authors need to justify why they chose the reported SD in this paper for their sample size calculation.

Response

Thanks for your comment! It is a very important and meaningful comment for our study. We have again read the references about CaTS, and found an article published in Lancet in 2018 (Lancet. 2018;392:1423-1433). The authors did an RCT and compared corticosteroid injection and night splints for CaTS. "6-week" was the primary endpoint in this study. Participants in the NS group (night splints) had a SD of 0.76 points for BCTQ-SSS at their 6-week followup. As the primary outcome of our study is also the 6-week BCTQ-SSS, therefore, we decide to choose 0.76 points as the SD value. After recalculating, a total of 48 participants in each group is required. Allowing for an up to 10% drop out rate, we aim to enroll at least 54 participants in each group to complete the study. At last, the final sample size is 54, not the original 21. Thanks for your kind comment again!

Changed text

Page 19, Line 419-429.

Sample size and power calculation are based on the primary outcome of BCTQ-SSS score at 6-week. Based on a published RCT trial ([Lancet. 2018;392:1423-1433], the authors compared corticosteroid injection and NS for CaTS, and participants in the NS group had a standard deviation [SD] of 0.76 points for BCTQ-SSS at their 6-week followup, ["6-week" was the primary endpoint in this study]), a SD of 0.76-point on BCTQ-SSS score will be used. a total of 48 participants in each group is required. Allowing for an up to 10% drop out rate, we aim to enroll at least 54 participants in each group to complete the study.

Comment 3

Is the primary outcome the symptom severity score at 3 months or change in score over time?

Response

Thanks for your comment! We are sorry for our previous ambiguous statement. The primary outcome was the symptom severity score at 6 weeks after treatment, nor the change in score or 3 months. "6-week" was also the primary endpoint in many other articles including splints, like [Lancet. 2018;392:1423-1433], which was usually used for 6 weeks.

Changed text

Page 14, Line 301.

The primary outcome measure will be the difference in Symptom Severity Scale of the Boston Carpal Tunnel Questionnaire (BCTQ-SSS) at 6-week.

Page 19, Line 419-420.

Sample size and power calculation are based on the primary outcome of BCTQ-SSS score at 6-week.

Comment 4

Considering that the trial has three groups and multiple follow-up measurements, there is no mention of how the issue of multiplicity will be managed.

Response

Thanks for your comment! It is very important and meaningful to improve our statistic analysis. There will be three possible comparisons from the three intervened groups. Bonferroni method will be used to adjust for multiplicity. Thanks for your kind advice again!

Changed text

Page 20, Line 441-442.

..... Bonferroni method will be used to adjust for multiplicity.

Comment 5

The authors chose several secondary outcomes and with this small sample size it would be unlikely that the study will have adequate power for the analysis of these secondary outcomes.

Response

Thanks for your comment! According to your kind comment 2, the sample size was recalculated. After recalculating, a total of 48 participants in each group is required. Allowing for an up to 10% drop out rate, we aim to enroll at least 54 participants in each group to complete the study now.

Comment 6

Abstract needs to include more information about the eligibility criteria and the primary outcome (timepoint and which groups will be compared).

Response

Thanks for your comment! We have rewritten the Abstract section, and add some important information according to your kind advice.

Changed text

Page 5, Line 99-105.

This study protocol entails a three-arm, prospective, multicenter, randomised controlled trial. 162 eligible adult participants diagnosed with mild to moderate CaTS by using criteria developed from a consensus survey by the UK Primary Care Rheumatology Society will be assigned to either (1) US, (2) night splint (NS) or (3) US+NS (combined) group. Primary outcome will be difference in Symptom Severity Scale of Boston Carpal Tunnel Questionnaire (BCTQ-SSS) at 6-week between NS and US+NS groups.

Thanks for your valuable time and comments again! The above comments are all valuable and very helpful for revising and improving our paper. We hope our revisions will meet your requirements.

Reviewer: 2

Dr. Philippe Paquette, University of Montreal

In general, the manuscript is well organised although several typographical and grammar errors are found. A thorough revision is required before publication. I recommend authors to hire a freelance writer proficient in English and academic work.

Response

Thanks for your kind comment! We have consulted a professional copyediting service to help us to edit the manuscript from beginning to end, and it seems more professional now. Thanks again for your kind advice!

Thanks for your valuable time and comments again! The above comments are all valuable and very helpful for revising and improving our paper. We hope our revisions will meet your requirements.

VERSION 2 – REVIEW

REVIEWER	Isam Atroshi Hösslholm Hospital, Department of Orthopedics
REVIEW RETURNED	30-Jan-2022
GENERAL COMMENTS	I have no further comments.

REVIEWER	Philippe Paquette University of Montreal, School of rehabilitation
REVIEW RETURNED	02-Feb-2022

GENERAL COMMENTS	A copy of this first paragraph will be forwarded to the editor. I thank the authors for their dedication to improving their manuscript as evidenced improvements in spelling and grammar. Unfortunately I could not determine if all my previous comments were answered point-by-point. In fact, I'm afraid the attached file that completed my initial review was missed by the authors. I hope they didn't consider that my comments lacked substance. Fortunately, a few points were touched upon by my colleague's work. I will therefore reiterate my previous unanswered comments with the addition of a major point regarding the portable therapeutic ultrasound machine.  • Portable ultrasound machine I was very surprised reading the authors response to my colleague's comment #1 to learn of the development of a portable ultrasound device for participants to use "everywhere, like home...". This considerably impacts the aim of this study. In fact, unless the authors could provide data from a previous pilot trial, I have serious concerns about the feasibility and safety of self-administered ultrasound treatments (see next comment). In addition, the results of this study would be difficult to compare with the literature, which mainly includes therapeutic ultrasounds treatments provided by health professionals. • Ultrasound therapy parameters Another main concern regards the choice to use continuous therapeutic ultrasound, which can cause overheating in the treatment area. Although the proposed parameters (i.e., 1 MHz 1.0 W/cm²; 15 min; 5 day/week) have been use in prior studies, adverse events have not been thoroughly documented (1). There are safety concerns in the use any type of therapeutic ultrasound which are outlined in Miller et al.(2) Furthermore, the authors should provide stronger arguments in favor of continuous mode, as studies referenced in the introduction highlight that pulsed ultrasound has effects on nerve conduction. • Secondary outcome measures The scope of secondary outcomes is unclear. The authors should consider that certain health domains to be evaluated overlap. For instance, functional status is measured in both the BCTQ and Quick-DASH. Likewise, pain and paresthesia are measured in both the BCTQ and with a NRS. Multiple patient reported questionnaires could potentially be burdensome for participants and reduce compliance. Conversely, the authors could consider including an economic or cost-effectiveness evaluation. Additionally, the relevance of EMG and ultrasound imaging outcomes should be clearly explained and perhaps re-think. The role of ultrasound imaging in the investigation of CTS is documented, however it has now been demonstrated as a responsive outcome to my knowledge. Did the author consider other ultrasound imaging measures such as echogenicity and texture? Miscellaneous:  • The common abbreviation of Carpal Tunnel Syndrome is CTS. The authors should provide a rational for using CaTS
--

	 • Please refer to 'therapeutic ultrasound' or 'ultrasound therapy' to avoid confusion with ultrasound imaging and avoid the 'US' abbreviation. • Line 136, typo 2.76% • Line 152, check for appropriate grammar. Consider using 'non-invasiveness'. • Line 231, typo inter-current • Line 238, inappropriate use of 'et al.' which should refer to a list of name. Also, I suggest to thoroughly list the contraindications. • Line 241, affected 'wrist' seems more appropriate References:  1. Page MJ, O'Connor D, Pitt V, Massy-Westropp N. Therapeutic ultrasound for carpal tunnel syndrome. Cochrane Database Syst Rev. 2013(3):CD009601. 2. Miller DL, Smith NB, Bailey MR, Czarnota GJ, Hynynen K, Makin IR, et al. Overview of therapeutic ultrasound applications and safety considerations. J Ultrasound Med. 2012;31(4):623-34.
--	--

VERSION 2 – AUTHOR RESPONSE

■ Reviewer: 1

Dr. Isam Atroshi, H^ässleholm Hospital

I have no further comments.

Response:

Thanks for your valuable time and comments again! The previous comments are all valuable and very helpful for revising and improving our paper.

Reviewer: 2

Dr. Philippe Paquette, University of Montreal

I thank the authors for their dedication to improving their manuscript as evidenced improvements in spelling and grammar. Unfortunately, I could not determine if all my previous comments were answered point-by-point. In fact, I'm afraid the attached file that completed my initial review was missed by the authors. I hope they didn't consider that my comments lacked substance. Fortunately, a few points were touched upon by my colleague's work. I will therefore reiterate my previous unanswered comments with the addition of a major point regarding the portable therapeutic ultrasound machine.

Response

Thanks for your kind comment! It's our big mistake that we actually miss the attached file from you in the first response, and only make response to reviewer 1. We are very sorry for it. And thanks for your advice again, that they are all valuable and very helpful for revising and improving our paper. The unanswered comments are finished below, including the major point regarding ultrasound machine, and we hope our revisions will meet your requirements. Thanks for your valuable time and comments again!

Comment 1

Portable ultrasound machine

I was very surprised reading the authors response to my colleague's comment #1 to learn of the development of a portable ultrasound device for participants to use "everywhere, like home...". This considerably impacts the aim of this study. In fact, unless the authors could provide data from a previous pilot trial, I have serious concerns about the feasibility and safety of self-administered ultrasound treatments (see next comment). In addition, the results of this study would be difficult to compare with the literature, which mainly includes therapeutic ultrasounds treatments provided by health professionals.

Response

Thanks for your kind comment! The portable therapeutic ultrasound machine is just being developed, not for used in this study. In our last response, what we want to show is that, if the results of therapeutic ultrasound in this study (still the conventional instrument used in hospital like other published literatures) are promising, we can deliver a further research on the portable therapeutic ultrasound (still in developing) machine in future. We are sorry for our unclear statement for misleading, and thank you again for your kind advice!

Comment 2

Ultrasound therapy parameters

Another main concern regards the choice to use continuous therapeutic ultrasound, which can cause overheating in the treatment area. Although the proposed parameters (i.e., 1 MHz 1.0 W/cm²; 15 min; 5 day/week) have been use in prior studies, adverse events have not been thoroughly documented (1). There are safety concerns in the use any type of therapeutic ultrasound which are outlined in Miller et al. (2) Furthermore, the authors should provide stronger arguments in favor of continuous mode, as studies referenced in the introduction highlight that pulsed ultrasound has effects on nerve conduction.

Response

Thanks for your kind comment! We are sorry in our previous manuscript for this big unaware written error. You are very careful and responsible, and much thanks again! (otherwise, it would be a big misleading to readers) Actually, the mode of therapeutic ultrasound in this study is pulsed (1:4), nor continuous, it must be a written error in our previous manuscript. We have referred to other papers at the design stage of this study, including [Ebenbichler GR, et al. BMJ. 1998;316:731-735]. Thank you again for your kind advice!

Change text

Page 13, line 273-277.

Participants in the [therapeutic ultrasound group] will receive pulsed therapeutic ultrasound (model 1:4, Shanghai, China) for 6 weeks at a frequency of 1 MHz and intensity of 1.0 W/cm² for 15 minutes per session, in daily 5 times a week for the first 2 weeks and twice a week for another 4 weeks, to the area over the carpal tunnel, referred to a published trial [BMJ. 1998;316:731-735].

Comment 3

Secondary outcome measures

The scope of secondary outcomes is unclear. The authors should consider that certain health domains to be evaluated overlap. For instance, functional status is measured in both the BCTQ and Quick-DASH. Likewise, pain and paresthesia are measured in both the BCTQ and with a NRS. Multiple patient reported questionnaires could potentially be burdensome for participants and reduce compliance. Conversely, the authors could consider including an economic or cost-effectiveness evaluation. Additionally, the relevance of EMG and ultrasound imaging outcomes should be clearly explained and perhaps re-think. The role of ultrasound imaging in the investigation of CTS is documented, however it has now been demonstrated as a responsive outcome to my knowledge. Did the author consider other ultrasound imaging measures such as echogenicity and texture?

Response

Thanks for your comment! It's a very nice advice, that we are not aware of the overlap evaluation, which will produce potential burdensome for participants and reduce compliance. After screening for our previous outcome measures, and combining with your suggestions, we find that, Quick-DASH and NRS are repeated with BCTQ in function and symptom evaluation. As BCTQ is the primary outcome in this study, we decided to delete the Quick-DASH and NRS in secondary outcomes. In addition, we also delete participant satisfaction, as we believe it is repeated with treatment success and recurrence rate (GROC on a six-level Likert scale) to some extent.

As for the ultrasound parameters used in this protocol, the cross-sectional area (CSA) of median nerve is an objective and quantitative measurement, which is more suitable for scientific research and

is one of the most used diagnostical and evaluated methods in clinical practice, as used in many other articles on CTS research, like [Ann Neurol. 2018;84:601-610] and [Mayo Clin Proc. 2017;92:1179-1189]. However, other ultrasound parameters such as echogenicity and texture might be qualitative (operator-dependent), nor quantitative, therefore, would not be suitable for scientific research. We believe CSA would be the best choice. Thanks for your kind advice again!

Deleted text

Page 15, line 320 and 324.

Comment 4

The common abbreviation of Carpal Tunnel Syndrome is CTS. The authors should provide a rationale for using CaTS.

Response

Thanks for your kind advice! We have changed “CaTS” to “CTS” thorough out the manuscript.

Comment 5

Please refer to ‘therapeutic ultrasound’ or ‘ultrasound therapy’ to avoid confusion with ultrasound imaging and avoid the ‘US’ abbreviation.

Response

Thanks for your kind advice! We have changed “US” to “therapeutic ultrasound” thorough out the manuscript, when it means a therapy by ultrasound. Meanwhile, we also changed all “NS” to “night splint”.

Comment 6

Line 137, typo 2.76%°

Response

Thanks for your kind advice! The data in sentence is actually 2.76‰ (276:100 000), as referred to [Ann Intern Med. 2015;163:ITC1].

Comment 7

Line 152, check for appropriate grammar. Consider using 'non-invasiveness'. Response

Thanks for your kind advice! It should be noun here. We have changed "non-invasive" to "non-invasiveness".

Changed text

Page 8, line 152

..... which has the advantages of low cost, high efficacy and non-invasiveness.

Comment 8

Line 233, typo inter-current.

Response

Thanks for your kind advice! We have changed "inter-current" to "current".

Changed text

Page 12, line 233

Current illness including,

Comment 9

Inappropriate use of 'et al.' which should refer to a list of names. Also, I suggest to thoroughly list the contraindications.

Response

Thanks for your kind advice! We have deleted the word "et al" to be more rigorous.

Deleted text

Page 12, line 240

Comment 10

Line 243, affected 'wrist' seems more appropriate.

Response

Thanks for your kind advice! The word “wrist” would be more appropriate, and we have changed it.

Changed text

Page 12, line 243

Demographic variables such as age, sex, body mass index, affected wrist (whether bilateral),
.....

Thanks for your valuable time and comments again! The above comments are all valuable and very helpful for revising and improving our paper. We hope our revisions will meet your requirements.

VERSION 3 – REVIEW

REVIEWER	Philippe Paquette University of Montreal, School of rehabilitation
REVIEW RETURNED	21-Mar-2022

GENERAL COMMENTS	The authors were diligent in their responses to my comments and provided satisfactory improvements to their manuscript. I will await the result of their study with fervor. One minor note pertains to the ultrasound imaging parameters. The authors can refer to these two studies for methods to quantitatively evaluate of echogenicity and texture of the median nerve, although I can agreed that cross-sectional area is currently the gold standard and therefor the most useful measure. • Impink BG, Gagnon D, Collinger JL, Boninger ML. Repeatability of ultrasonographic median nerve measures. Muscle Nerve. 2010 Jun;41(6):767-73. doi: 10.1002/mus.21619. PMID: 20513104• Paquette P, Higgins J, Gagnon DH. Peripheral and Central Adaptations After a Median Nerve Neuromobilization Program Completed by Individuals With Carpal Tunnel Syndrome: An Exploratory Mechanistic Study Using Musculoskeletal Ultrasound Imaging and Transcranial Magnetic Stimulation. J Manipulative Physiol Ther. 2020 Jul-Aug;43(6):566-578. doi: 10.1016/j.jmpt.2019.10.007. Epub 2020 Aug 26. PMID: 32861518. I have no further comments.
--